# Automated Multi-Task Learning for Joint Disease Prediction on Electronic Health Records

**Suhan Cui**
The Pennsylvania State University
University Park, PA, USA
suhan@psu.edu

**Prasenjit Mitra**
The Pennsylvania State University
University Park, PA, USA
pmitra@psu.edu

## Abstract

Electronic Health Records (EHR) have become a rich source of information with the potential to improve patient care and medical research. In recent years, machine learning models have proliferated for analyzing EHR data to predict patients' future health conditions. Among them, some studies advocate for multi-task learning (MTL) to jointly predict multiple target diseases for improving the prediction performance over single task learning. Nevertheless, current MTL frameworks for EHR data have significant limitations due to their heavy reliance on human experts to identify task groups for joint training and design model architectures. To reduce human intervention and improve the framework design, we propose an automated approach named `AutoDP`, which can search for the optimal configuration of task grouping and architectures simultaneously. To tackle the vast joint search space encompassing task combinations and architectures, we employ surrogate model-based optimization, enabling us to efficiently discover the optimal solution. Experimental results on real-world EHR data demonstrate the efficacy of the proposed `AutoDP` framework. It achieves significant performance improvements over both hand-crafted and automated state-of-the-art methods, also maintains a feasible search cost at the same time. Source code can be found via the link: `https://github.com/SH-Src/AutoDP`.

## 1 Introduction

In the era of big data and digital healthcare, the voluminous Electronic Health Records (EHR) can revolutionize patient care, medical research, and clinical decision-making. Using these, the machine learning (ML) community has been designing models to predict patients' future health conditions, e.g., models for mortality prediction [1], diagnosis prediction [2, 3] and hospital readmission [4]. Although most existing machine learning based prediction models are designed to be single-task, i.e. predicting the risk of a single target disease, some works [5, 6, 7, 8, 9] designed multi-task learning (MTL) models to jointly predict multiple targets. The motivation lies in the fact that two or more diseases might be related to each other in terms of sharing common comorbidities, symptoms, risk factors, etc. Consequently, training on related diseases simultaneously can offer additional insights and potentially enhance prediction performance. While multi-task learning offers potential advantages, the existing MTL frameworks for EHR data still suffer from the following limitations.

**Limitations of the existing MTL frameworks for EHR data**. To design an effective MTL framework, two fundamental challenges need to be addressed:

*(1) How can we determine which tasks should be trained together?* The task grouping problem [10] involves finding groups of tasks to train jointly. Multi-task learning only provides advantages when the tasks are synergistic, i.e., training on the tasks together makes the model learn general knowledge that helps in performing the tasks better in the test set and prevents overfitting. Thus, given a large set

of related tasks in a domain, we may need to group the tasks (allowing tasks to belong to multiple groups) together to create groups of tasks on each of which we will train a model. However, existing works usually rely on human expert discretion to select multiple tasks upfront and create a shared model for those tasks [5, 6, 7, 8, 9]. Hence, none of them has addressed the general problem of task grouping for EHR data. Moreover, due to the complexity of disease correlations, grouping synergistic tasks together is extremely challenging for human experts. It not only demands substantial effort (trying out every possible task combination) but also introduces the risk of task interference (putting disparate diseases together), potentially leading to performance degradation. Therefore, how to design the appropriate task grouping for MTL on EHR data presents a critical challenge.

*(2) How can we design model architectures for MTL?* Existing works [5, 6, 7, 8, 9] typically rely on hand-crafted architectures for multi-task learning, which consist of a shared EHR encoder followed by several task-specific classifiers. However, due to the large number of possible operations as well as network topologies, manually tuning an optimal architecture for MTL is impossible. Furthermore, the optimal architectures for different task groups might also be distinct. Thus, things can even get worse when the number of tasks grows and different task combinations are involved for joint tuning. Therefore, we need a more efficient and effective approach to design the optimal MTL architectures for EHR data.

**Automating the MTL framework design for EHR data**. To address the aforementioned challenges, we look to Automated Machine Learning (AutoML) [11]. Since AutoML relies on data-driven approaches to automate the design of machine learning algorithms, it has the potential to improve the design of an MTL framework for EHR data and reduce human interventions. Several attempts have been explored in other domains, e.g., computer vision, to improve the design of task grouping [12, 10, 13] and MTL architectures [14, 15, 16, 17, 18]. However, the exploration of AutoML in healthcare domain remains relatively limited [19]. To the best of our knowledge, there are no existing work that automates the finding of groups of tasks for MTL towards designing an optimal framework for classification tasks using EHR data, which is a notable gap in the field.

**Joint optimization over task grouping and architecture search**. Morever, currently there exists no end-to-end optimization framework for automating MTL, even in other domains. Current approaches independently address the problems of task grouping and architecture design. First, a line of work [12, 10, 13] solves the task grouping problems by learning the task correlations. They operate under the underlying assumption that MTL architectures are the same across different task groups, which might not be practical nor optimal. Second, researchers also apply Neural Architecture Search (NAS) [20] to automatically design MTL architectures for a predefined set of tasks [14, 15, 16, 17, 18, 21]. No existing work has integrated these two approaches to address both problems simultaneously. However, combining them naively could lead to sub-optimal results, as sequential optimization might result in inaccurate estimations for both aspects. Therefore, we need a more generalized AutoML framework for the joint optimization of both task grouping and architecture search.

**Overview of the proposed approach**. Therefore, in this paper, we show that an integrated approach for multi-task grouping and neural architecture search provides significant improvements. First, we extend existing single-task models like Retain [22], Adacare [16] to MTL in an EHR setting. Second, we apply DARTS [23], an NAS method used in MTL settings in different domains to the EHR domain. We use one shared model for predicting multiple tasks. These adaptations improve over the single-task setting. Second, we explore the impact of automated task grouping in the EHR setting by grouping tasks and finding an optimal NAS model for each task group. This further improves the performance. Finally, we propose an integrated framework an **Auto**mated multi-task learning framework, **AutoDP**, for joint **D**isease **P**rediction on electronic health records, which aims at jointly searching for the optimal task grouping and the corresponding neural architectures that maximize the multi-task performance gain. We show that this third method provides the maximal performance gain.

Specifically, in **AutoDP**, we employ a surrogate model-based optimization approach [24] for efficient search. First, we define the joint search space of task combinations and architectures that includes all possible configurations for MTL. We want to find optimal solutions from this search space. To achieve that, the first question is how we can evaluate the performance of each configuration. Performing the ground truth evaluation for every configuration is infeasible, since it requires an entire multi-task learning procedure for each pair of architecture and task combination. Therefore, instead of exhaustively evaluating all the configurations, we build a surrogate model to predict the multi-task

gains for any given configurations from the search space. In this way, we only need to evaluate the ground truth gains for a subset of samples from the search space, and use them to train the surrogate model for estimating the rest ones. The intuition is that there exists an underlying mapping from each configuration to the expected multi-task gains; thus it can be learned by a neural network. The remaining question is how we can effectively train the surrogate model using as few samples as possible. To this end, we further propose a progressive sampling strategy to guide the surrogate model training for improving sample efficiency. That is we train the surrogate model through multiple iterations. At every iteration, we select some points from the search space and update the surrogate model accordingly. The selection is conditioned on the current surrogate model and involves both exploitation and exploration. That is, we iteratively select the points that bring higher performance gains and also come from unexplored areas, which makes the training samples represent the whole search space. Eventually, after we obtain the trained surrogate model, we further use it to derive the final optimal task grouping and architectures. Because of the huge search space, it is not practical to use brute-force search. Hence, we develop a greedy search method to find the near-optimal solution.

In summary, our contributions are as follows:

- We are the first to propose an automated approach for multi-task learning on electronic health records **AutoDP**, which largely improves the design of task grouping and model architectures by reducing human interventions. Specifically, this work is the first to automate the design for the optimal task grouping and model architectures for MTL on EHR data.

- We are the first to propose a surrogate model based optimization framework that jointly searches for the optimal task grouping and corresponding model architectures with high efficiency in any domain.

- We propose a progressive sampling strategy to construct the training set for the surrogate model, which improves sample efficiency by reducing the required number of ground truth evaluations during searching. Importantly, we balance exploitation and exploration so that the sampled configurations can represent the whole search space and are highly accurate.

- We propose a greedy search algorithm to derive the final MTL configuration using the trained surrogate model and find a near-optimal solution from the huge search space efficiently.

- Experimental results on real world EHR data - MIMIC IV [25] demonstrate that **AutoDP** improves classification performance significantly over existing hand-crafted and automated methods under feasible computational costs.

## 2 Related Work

**Multi-Task Learning with EHR**. To enhance prediction performance while forecasting patients' health conditions based on their historical data [26], existing studies employ multi-task learning to simultaneously predict multiple related target diseases or conditions, resulting in improved performance compared to single-task training. For example, Wang, et al. [7] investigated the advantages of joint disease prediction using traditional machine learning models. More recently, researchers have applied recurrent neural network (RNN) based models to conduct multi-task learning on EHR data [27, 6, 5], which is able to predict tasks like mortality, length of stay, ICD-9[1] diagnoses and etc. Additionally, Zhao, et al. [8] also utilized a transformer based method for multi-task clinical risk prediction on multi-modal EHR data. However, all these studies manually select the set of tasks for joint training without task grouping and utilize a hand-crafted MTL model architecture, which largely limits their performance.

**Multi-Task Grouping**. Due to the limitation of manually selected task groups, some of the work focus on obtaining the optimal task grouping through searching. Specifically, Standley, et al. [10] is the first work that systematically analyze the task correlations. For improving the efficiency, they use pair-wise MTL gains to estimate the high-order MTL gains, and obtain the pair-wise gains by training one model for each task pair. Based on the estimated gains, they derive the optimal task grouping using brute-force search. Fifty, et al. [12] further improves the efficiency by training one model to derive all the pair-wise gains. They derive the task affinity based on the gradient information during training. Furthermore, Song, et al. [13] propose a more general method that employs a meta model to

---

[1]https://www.cdc.gov/nchs/icd/icd9.htm

learn the task correlations and estimates the high-order MTL gains more effectively. These works normally assume that the model architecture is the same across different task groups. But in practice, we can maximize performance gains by applying different model architectures with respect to each task group. Thus, we need a more general framework that considers the model architectures during task grouping.

**Multi-Task NAS**. Neural Architecture Search (NAS) [20] stands as a prominent research area in AutoML, focusing on the exploration of optimal deep network architectures through a data-driven approach. Although the main stream of NAS focuses on the setting of single task learning, some researchers also try to employ NAS in multi-task learning applications, predominantly for searching computer vision MTL architectures. Notably, studies done by Ahn, et al. [14] and Bragman, et al. [15] employ reinforcement learning and variational inference, respectively, to determine whether each filter in convolutions should be shared across tasks. Furthermore, other recent works [16, 17, 18, 21] leverage differentiable search algorithms [23], to determine the optimal sharing patterns across multiple network layers for diverse tasks. Despite the demonstrated advancements, a common limitation is their reliance on human experts to pre-define a set of tasks for joint training. This constraint poses challenges in practical scenarios where task grouping is not readily available, thereby limiting their broader applicability. What is more, their frameworks often search for better MTL architectures on top of one or several backbone architectures such as ResNet [28]. However, such backbone architectures might not be available for EHR applications in medical domain. Therefore, a new multi-task NAS framework is needed for EHR data.

# 3 Methodology

## 3.1 Preliminaries

**Problem definition**. Assume we have the input EHR data for multiple patients where each patient is represented as $\mathbf{X} \in \mathbb{R}^{L \times d_e}$, where $L$ is the time sequence length and $d_e$ is the hidden dimension of the input features. We have $N$ prediction tasks using the EHR data, denoted as $\mathcal{T} = \{T_1, T_2, \cdots, T_N\}$. We seek to maximize the overall MTL performance gain for all these prediction tasks compared to single task training. First, we define MTL gain. Conduct a single task training on each task independently using a specific backbone model (such as RNN), and obtain the single-task performance for all tasks in terms of a predefined metric (such as average precision), denoted as $\{s_1, s_2, \cdots, s_N\}$. Then, the MTL gain is defined as:

$$g_i = \frac{(m_i - s_i)}{s_i}, i = 1, \cdots, N, \tag{1}$$

where $m_i$ is the multi-task performance for $T_i$. Therefore, our objective is to maximize the overall gain for all tasks: $G = \frac{1}{N} \sum_{i=1}^{N} g_i$.

To achieve that, our proposed method solves two searching problems at the same time using AutoML. First, we search for a list of task combinations that defines which tasks should be trained together. Second, we determine the optimal model architecture for each task combination. We aim at finding the optimal configuration for both, such that the highest overall gain $G$ is attained.

**Task grouping search space**. For $N$ tasks, there are $2^N - 1$ task combinations, $\mathcal{C} = \{C_1, C_2, \cdots, C_{2^N-1}\}$, where every $C$ is a subset of $\mathcal{T}$. Given a budget $B$, we aim at searching for maximally $B$ task combinations from $\mathcal{C}$ to determine which tasks should be trained together. The task combinations should cover all $N$ tasks so that we are able to obtain $\{m_1, m_2, \cdots, m_N\}$. If one task $T_n$ appears in multiple task combinations, we simply choose the highest performance for it as $m_n$.

**Architecture search space**. For every task combination, we also need to search for an MTL architecture to model the EHR data. We adopt the hard sharing mechanism as in most existing works [27, 5], which consists of a shared encoder for extracting the latent representation of the input EHR and multiple task specific classifiers to generate the output for every task.

Specifically, we enable the search for the optimal shared encoder. For the search space of the encoder, we adopt the setting of directed acyclic graph (DAG) [23]. The architecture is represented as a DAG that consists of $P$ ordered computation nodes, and each node is a latent feature that has connections to all previous nodes. For each connection (also called edge), we can choose one operation from a

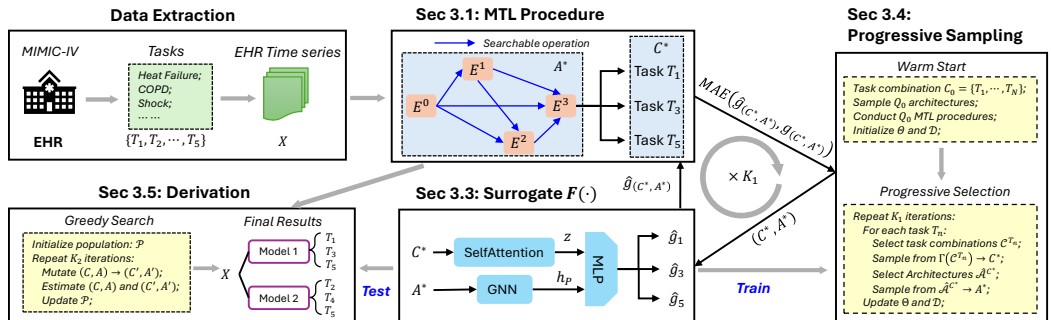

Figure 1: Overview of the proposed `AutoDP`

predefined set of candidate operations $\mathcal{O}$ for feature transformation. Let $\mathbf{E}^0 = \mathbf{X}$, the formulation of node $p$ is defined as follows:

$$\mathbf{E}^p = \sum_{i=0}^{p-1} o_{(i,p)}(\mathbf{E}^i), o_{(i,p)} \in \mathcal{O}, \tag{2}$$

where node features $\mathbf{E}^i \in \mathbb{R}^{L \times d_e}$'s all have the same dimension as $\mathbf{X}$, and $o_{(i,p)}$ is the operation that transform $\mathbf{E}^i$ to $\mathbf{E}^p$. Essentially, sampling one architecture from the search space is equivalent to sampling one operation for every edge in the DAG. In this way, we can get the set of all possible architectures denoted as $\mathcal{A}$.

Finally, to predict, we take the last node representation $\mathbf{E}^P$ as the encoded feature for the input EHR, and use task-specific classifiers to output final predictions, which are all fixed fully connected network layers.

**MTL procedure**. To evaluate a specific sample from the joint search space $\mathcal{C} \times \mathcal{A}$, we need to conduct an MTL experiment to obtain the multi-task performances. Specifically, given an architecture $A \in \mathcal{A}$ and a task combination $C \in \mathcal{C}$, we train the model $A$ to predict for all tasks in $C$ and get the multi-task performances for those tasks. Then, we can compute their gains by Eq.(1). In this way, we are able to evaluate how much gains that this sample $(C, A)$ could achieve.

## 3.2 Overview

We propose a surrogate model based AutoML framework to search for the optimal task grouping and corresponding architectures simultaneously. To achieve that, we need to first evaluate the MTL gains for all the samples in the joint search space $\mathcal{C} \times \mathcal{A}$, and then select the best $B$ samples (pairs of task combinations and architectures) that maximize $G$. However, it is not practical to obtain the ground-truth gains for every sample, since the whole search space is normally very huge and every MTL procedure is also considerably expensive. Therefore, we build a neural network (called surrogate model) to learn the mapping from a pair of task combination and architecture to the multi-task gains:

$$\mathbf{g}_{(C,A)} = F(C, A), C \in \mathcal{C}, A \in \mathcal{A}, \tag{3}$$

where $\mathbf{g}_{(C,A)} \in \mathbb{R}^{|C|}$ is the per-task gains for task combination $C \in \mathcal{C}$ if using $A$ as the model, and $F(\cdot)$ is the surrogate model. In this way, we only need to evaluate the ground truth gains for a small subset of samples from the search space, and use them to train the surrogate model for estimating all other unseen samples. The assumption is that the multi-task gains are essentially determined by the configuration of the task combination and the architecture, so there exists an underlying mapping that could be learned by a neural network. We set universal hyperparameters and optimization settings for all MTL procedures, hence the influence of other factors can be ignored.

Specifically, we introduce the model architecture of the surrogate model in Section 3.3. Then, we outline the training procedure of the surrogate model in Section 3.4, where we propose an active learning strategy to collect training samples. Eventually, we use greedy search to derive the final configuration of task grouping and architectures by utilizing the trained surrogate model, as discussed in Section 3.5. The framework overview is shown in Figure 1.

### 3.3 Surrogate Model

For learning the mapping from an input configuration to the multi-task gains, the surrogate model is required to encode both architectures and task combinations. Also, the model needs to output multi-task gains. Therefore, we design a new surrogate model that consists of two encoders that respectively transform the input architecture and task combination into latent representations. Then, two representations are fused together to predict the multi-task gains.

**Architecture Encoding**. For encoding a given architecture $A$, we apply a graph encoder [29] that is specifically designed for modeling DAGs, which is suitable for encoding the architectures in our search space. It can sequentially update the hidden states for the $P$ computation nodes in preceding order by aggregating information from all predecessors. For node $p$, we have:

$$\mathbf{h}_p = \text{Aggregate}(\mathbf{W}_0 \cdot \mathbf{h}_0, \mathbf{W}_1 \cdot \mathbf{h}_1, \cdots, \mathbf{W}_{p-1} \cdot \mathbf{h}_{p-1}), \tag{4}$$

where $\mathbf{h}_0 \in \mathbb{R}^{d_s}$ is the input node representation which contains trainable parameters, and $\mathbf{W} \in \mathbb{R}^{d_s \times d_s}$'s are learnable transition matrices constructed for each operation in $\mathcal{O}$. For every operation in the architecture, we also apply the corresponding $\mathbf{W}$ in our graph encoder. For aggregating all incoming representations, we apply average pooling to obtain the node representation $\mathbf{h}_p$. Finally, we use the node representation for the last node $\mathbf{h}_P$ as the overall encoding for the input architecture.

**Task Combination Encoding**. For encoding a given task combination $C$, we use the self attention mechanism [30] to model the high order interactions among the selected tasks in $C$. Specifically, we randomly initialize the embedding for all $N$ tasks, and for task combination $C$, we have:

$$\mathbf{z} = \text{Pool}(\text{SelfAttention}(\mathbf{u}_1, \mathbf{u}_2, \cdots, \mathbf{u}_{|C|})), \tag{5}$$

where $\mathbf{u} \in \mathbb{R}^{d_s}$'s are corresponding embeddings for the selected tasks and $\mathbf{z} \in \mathbb{R}^{d_s}$ is the final representation for task combination $C$. Additionally, we also use average pooling on top of the self attention layers to obatin $\mathbf{z}$.

**Prediction**. Eventually, we apply a two layer MLP to fuse both architecture encoding $\mathbf{h}_P$ and task combination encoding $\mathbf{z}$, and output the predicted gains for all selected tasks $\hat{\mathbf{g}}_{(C,A)} \in \mathbb{R}^{|C|}$. We use the mean absolute error to supervise the surrogate model as follows:

$$\mathcal{L}(\hat{\mathbf{g}}_{(C,A)}, \mathbf{g}_{(C,A)}) = ||\hat{\mathbf{g}}_{(C,A)} - \mathbf{g}_{(C,A)}||_1, \tag{6}$$

where $\mathbf{g}_{(C,A)} \in \mathbb{R}^{|C|}$ is the ground truth gains generated by conducting an MTL procedure for $(C, A)$.

### 3.4 Progressive Sampling

In order to efficiently train the surrogate model defined in previous section, we develop a progressive sampling method to collect training samples. Start with an empty training set and a random initialized surrogate model, we progressively sample more points from the search space $\mathcal{C} \times \mathcal{A}$, and then use them to train the surrogate model. Specifically, we include two stages for the surrogate model training:

**Warm start**. Firstly, we warmup the surrogate model by selecting a small number of samples from the search space. Specifically, we use the task combination that contains all $N$ tasks $C_0 = \{T_1, \cdots, T_n\}$ and randomly sample $Q_0$ architectures from $\mathcal{A}$. Then we conduct $Q_0$ MTL procedures to evaluate their gains by training on $C_0$. In this way, we collect $Q_0$ training samples as the initial training set denoted as $\mathcal{D}$. We further train the surrogate model on $D$, and denote the model parameters as $\mathbf{\Theta}$.

**Progressive selection**. Then, we progressively select more points and train the surrogate model as introduced in Algorithm 1. Totally, we conduct $K_1$ rounds of sampling. For each round, we iterate through all $N$ tasks. With respect to one task $T_n$, we build the acquisition function $\mathbf{\Gamma}$ over the set of task combinations that contains $T_n$ based on the predicted gains for $T_n$. Then, we select one task combination $C^*$ that have highest value. We apply Upper Confidence Bound [31] as the acquisition function that considers both exploration and exploitation by explicitly estimating the mean and variance of predicted gains (line 11 marked by blue). Besides that, we would also like to see the effect of exploration vs exploitation. so we try out different settings of $\mathbf{\Gamma}$. Specifically, we propose three variants of `AutoDP`, namely `AutoDP`$^{\mu+\sigma}$, `AutoDP`$^{\mu}$ and `AutoDP`$^{\sigma}$, which corresponds to the original setting, including only $\mu$ or only $\sigma$ in $\mathbf{\Gamma}$. In this way, we can compare the results with pure exploration and pure exploitation during sampling, and find out the optimal strategy for `AutoDP`.

---

**Algorithm 1:** Progressive Selection

**Input:** Training set $\mathcal{D}$, surrogate model parameter $\Theta$, $Q_1, Q_2, K_1$; $Q_1 > Q_2$.
**Output:** Updated $\mathcal{D}$ and $\Theta$

**1 for** $k = 1, 2, \cdots, K_1$ **do**
**2**     **for** $n = 1, 2, \cdots, N$ **do**
**3**        Collect all task combinations that contains $T_n$: $\mathcal{C}^{T_n} = \{C_j | \forall C_j \in \mathcal{C}, T_n \in C_j\}$;
**4**        **for** $\forall C_j \in \mathcal{C}^{T_n}$ **do**
**5**           Randomly sample $Q_1$ architectures from $\mathcal{A}$, denote the set as $\mathcal{A}^{C_j}$;
**6**           Forward the surrogate model to collect gains for $T_n$ with every architecture in $\mathcal{A}^{C_j}$:
            $\mathcal{G} = \{\mathbf{g}[T_n] | \forall A \in \mathcal{A}^{C_j}, \mathbf{g} = F(C_j, A)\}$;
**7**           Select the top $Q_2$ architectures from $\mathcal{A}^{C_j}$ with highest gains in $\mathcal{G}$, denoted as $\hat{\mathcal{A}}^{C_j}$;
**8**           Calculate the mean over top $Q_2$ gains from $\mathcal{G}$, denoted as $\mu^{C_j}$;
**9**           Calculate the variance over top $Q_2$ gains from $\mathcal{G}$, denoted as $\sigma^{C_j}$;
**10**        **end**
**11**        Compute the acquisition values over $\mathcal{C}^{T_n}$ as: $\mathbf{\Gamma}(\mathcal{C}^{T_n}) = \{\mu^{C_j} + \lambda \cdot \sigma^{C_j}, \forall C_j \in \mathcal{C}^{T_n}\}$;
**12**        Sample a task combination $C^*$ from $\mathcal{C}^{T_n}$ that has highest value in $\mathbf{\Gamma}(\mathcal{C}^{T_n})$, and randomly
         sample an architecture $A^*$ from $\hat{\mathcal{A}}^{C^*}$;
**13**        Conduct an MTL procedure on $(C^*, A^*)$, and collect the ground truth labels $\mathbf{g}_{(C^*, A^*)}$;
**14**        Add $(C^*, A^*, \mathbf{g}_{(C^*, A^*)})$ to $\mathcal{D}$;
**15**     **end**
**16**     Update $\Theta$ by training the surrogate model on $\mathcal{D}$;
**17 end**

---

Moreover, we also select one architecture $A^*$ with high predicted gain for $T_n$ when combined with $C^*$. The selection of $C^*$ and $A^*$ is interdependent, and the details are introduced in Algorithm 1. In this way, we collect one sample $(C^*, A^*)$ to update the training set $\mathcal{D}$ with respect to each $T_n$. At the end of each round, we also update the surrogate model parameters $\Theta$ with the updated $\mathcal{D}$. After $K_1$ rounds, we are able to obtain a well trained surrogate model for estimating the whole search space.

### 3.5 Derivation

We derive the final results using the trained surrogate model. Due to the huge search space, it is still not practical to use brute force search to get the global optimum. Therefore, we propose to apply a greedy method to search for near-optimal solutions. We introduce the detailed procedure in Algorithm 2. The high level idea is that we first randomly initialize the configuration, and then gradually improve its multi-task gain by random mutation and greedy selection.

Specifically, given the budget $B$, we aim at searching for $B$ samples from the search space $\mathcal{C} \times \mathcal{A}$ such that the overall gain $G$ is maximized. We first randomly initialize the population $\mathcal{P}$ that contains $B$ pairs of task combinations and architectures. Then, at every iteration, we randomly mutate one pair $(C, A)$ from the population and see whether the overall multi-task gain will increase. If so, we

---

**Algorithm 2:** Greedy Search

**Input:** Trained surrogate model $F(\cdot)$, $B$, $K_2$.
**Output:** Searched population $\mathcal{P}$.
**1** Randomly sample $B$ pairs from $\mathcal{C} \times \mathcal{A}$ to initialize $\mathcal{P}$ ;
**2 for** $v = 1, 2, \cdots, K_2$ **do**
**3**     Randomly select one pair $(C, A)$ from $\mathcal{P}$;
**4**     Mutate $(C, A)$ to $(C', A')$ by changing one task in $C$ or one operation in $A$, and obtain a new
      population $\mathcal{P}'$;
**5**     Estimate $\mathcal{P}'$ and $\mathcal{P}$ using $F(\cdot)$;
**6**     Choose the better one: $\mathcal{P} \leftarrow Select(\mathcal{P}', \mathcal{P})$ ;
**7 end**

---

update $\mathcal{P}$ accordingly. After $K_2$ iterations, we can obtain a near-optimal solution. In practice, we also apply multiple initial populations to avoid getting stuck on local optima. Although we only get an approximate solution, our method can already achieve significant improvements over baselines as shown in Section 4.2.

# 4 Experiments

## 4.1 Set Up

**Dataset & Tasks**. We adopt MIMIC - IV dataset [25] for our experiments, which is a publicly available database sourced from the electronic health record of the Beth Israel Deaconess Medical Center. Specifically, we extract the clinical time series data for the 56,908 ICU stays from the database as our input EHR data, with an average sequence length of 72.9. With respect to each ICU stay, we also extract **25 prediction tasks** (listed in Table 3), including chronic, mixed, and acute care conditions. Each condition is associated with a binary label indicating whether the patient has the corresponding condition during the ICU stay.

**Baselines**. To compare the proposed method with existing work, we choose several state-of-art-baselines, including both *hand-crafted* and *automated* methods. Specifically, as described below, we include several human-designed EHR encoders to compare with the searched architecture we defined in Eq. (2). Also, we include one NAS method and one multi-task grouping method as the automated baselines. More importantly, we combine the multi-task grouping method with the NAS method and hand-crafted encoders to show the superiority of our joint optimization method.

- EHR encoders: We choose four models that are widely utilized for analyzing EHR time series, including LSTM [32], Transformer [30], Retain [22] and Adacare [1].

- NAS: We choose DARTS [23] as the NAS baseline, which is a differentiable search method for efficient architecture search. We apply it to our search space $\mathcal{A}$ to find better EHR encoders. Several state-of-the-art works in other domains have also used it to find MTL architectures [16, 17, 18, 21].

- Multi-task grouping: MTG-Net [13] is the current state-of-the-art multi-task grouping algorithm, which uses a meta learning approach to learn the high-order relationships among different tasks. We refer to this method as MTG in latter sections.

**Evaluation Metric**. We use two widely used metrics for binary classification to evaluate our method and baselines: **ROC** (Area Under the Receiver Operating Characteristic curve) and **AVP** (Averaged Precision). During surrogate model training, we use **AVP** as the metric to compute multi-task gains as in Eq. (1), since it is a more suitable choice for considering the class imbalance.

*Please also refer to Appendix A for the implementation details.*

## 4.2 Performance Evaluation

We show our results in Table 1. Each experiment is run five times and the average of the runs are reported. We run three settings: **Task @ 5**, **Task @ 10** and **Task @ 25**, which refers to using the first 5 tasks, 10 tasks and 25 tasks respectively. Since grouping all 25 tasks takes a long time to run, we include two small settings that only have the first 5 or 10 tasks in Table 3 for grouping. Our results demonstrate our hypotheses: (a) applying Retain, Adacare, and DARTS improves over the single-task setting, (b) applying different NAS models for each group further improves the performance, and finally, (c) `AutoDP` provides the best results in terms of averaged per-task gain for **ROC** and **AVP**, a significant improvement over existing MTL frameworks for EHR data.

First, without considering task grouping, we train one shared model to predict for all tasks in three settings and compute the multi-task gains for them. Results show that this setting only provides minimal improvement over single task training. Note that the automated method (DARTS) performs better than other hand-crafted methods. We also see that sequential optimization over task grouping and architecture search (MTG+DARTS) performs better than MTG + other hand-crafted encoders.

Moreover, we see that the three variants of `AutoDP` performed better than the other methods. Among them, `AutoDP`$^{\mu+\sigma}$ performs the best, which means the balance of exploration and exploitation is the most effective strategy for training the surrogate model. For the last method, we also report the standard deviations and p-values of statistical tests (compared to MTG+DARTS), which justifies that

Table 1: Performance comparison in terms of averaged per-task gain over single task backbone (All results are in the form of percentage values %).

| Settings | Included Tasks | Tasks @ 5 | | Tasks @ 10 | | Tasks @ 25 | |
|---|---|---|---|---|---|---|---|
| | Metric | ROC | AVP | ROC | AVP | ROC | AVP |
| One model for all tasks | LSTM | +0.09 | +0.18 | +1.06 | +3.22 | +1.83 | +7.46 |
| | Transformer | +0.97 | +4.82 | +1.41 | +4.14 | +1.75 | +7.45 |
| | Retain | +0.46 | +1.80 | +0.66 | +0.75 | +1.41 | +5.88 |
| | Adacare | +1.03 | +5.21 | +1.32 | +4.05 | +1.68 | +6.94 |
| | DARTS | +1.28 | +5.01 | +2.01 | +6.87 | +1.87 | +7.71 |
| Task Grouping + One model for each group | MTG+LSTM | +0.51 | +2.10 | +0.65 | +1.87 | +1.74 | +7.40 |
| | MTG+Transformer | +0.91 | +3.64 | +1.20 | +3.95 | +1.79 | +9.15 |
| | MTG+Retain | +0.55 | +3.11 | +1.51 | +5.20 | +1.54 | +8.87 |
| | MTG+Adacare | +1.25 | +5.78 | +1.44 | +4.63 | +1.75 | +7.84 |
| | MTG+DARTS | +1.47 | +6.41 | +2.02 | +6.65 | +2.41 | +11.76 |
| Variants of **AutoDP** | **AutoDP**$^{\mu}$ | +1.49 | +7.12 | +2.08 | +7.53 | +2.68 | +12.70 |
| | **AutoDP**$^{\sigma}$ | **+1.95** | +7.68 | +2.49 | +8.45 | +2.62 | +13.37 |
| | **AutoDP**$^{\mu+\sigma}$ | +1.69 | **+7.74** | **+2.55** | **+8.81** | **+2.80** | **+13.43** |
| | (std) | ± 0.08 | ± 0.25 | ± 0.13 | ± 0.29 | ± 0.12 | ± 0.33 |
| | (p-value) | 0.045 | 0.029 | 0.036 | 0.045 | 0.027 | 0.032 |

the improvement is significant. *The runtime is approximately as the same for MTG+DARTS and **AutoDP** and thus this is a fair comparison.*

Beside the overall performance gain, we also look at the distribution of performance gains for each individual task as shown in Figure 2. We can observe that the proposed method does not have the issue of negative transfer, since all tasks have a positive gain. Also, for some of the tasks, it can achieve over $20\%$ improvement, which further shows the effectiveness of **AutoDP**.

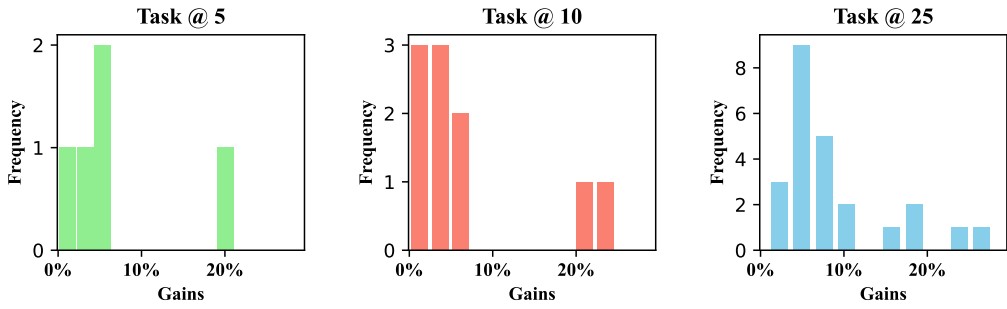

Figure 2: Histogram of task gains for **AutoDP** in terms of Averaged Precision.

### 4.3 Hyperparameter & Complexity Analysis

We analyze the effect of two vital hyperparameters of our method: $K_1$ and $B$, since they are the crucial parameters that largely define the complexity of our method during searching and inference respectively. We choose the setting of **Task @ 25** for a comprehensive analysis of all tasks. We try out different values and report the corresponding performance gain (**AVP**) in Figure 3.

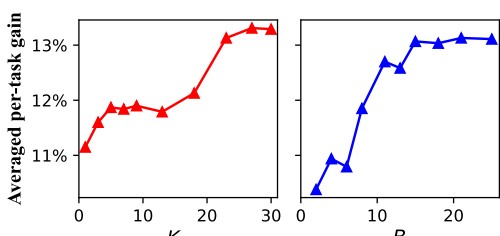

Figure 3: Analysis for the number of progressive sampling rounds $K_1$ and the budget of task groups $B$ under the setting of Task @ 25.

First, $K_1$ determines the number of training samples collected during searching. Given that each sample invokes an MTL procedure, it constitutes the major portion of the search cost. Therefore, our goal is to find an optimal value for $K_1$, striking a balance between cost-effectiveness and achieving commendable performance. We notice that the performance change plateaus after $K_1$ reaches 25.

That is, the surrogate model effectively learns the distribution of the search space after consuming $25 \times 25$ training samples during active selection (25 samples per round). Consequently, we can empirically decide to halt the iteration at this point.

Second, $B$ determines the number of task groups for the final configuration, which indicates the number of MTL models required for achieving the expected performance gain after searching. We also observe similar phenomenon that the performance becomes stable after $B$ reaches 12. We could also choose the optimal value for $B$ accordingly.

### 4.4 Ablation Study

We further analyze the effect of several components within `AutoDP`, including progressive sampling (Section 3.4), greedy search (Section 3.5), and task grouping as a whole. We replace these components with naive or human intuition-inspired baselines and report the performances in Table 2. Removing any of the components from the original framework leads to noticeable performance decreases, demonstrating the effectiveness of the designed components.

Table 2: Ablation results in terms of **AVP**.

| Settings | Task @ 5 | Task @ 10 | Task @ 25 |
|---|---|---|---|
| `AutoDP` | **+7.74** | **+8.81** | **+13.43** |
| Random sampling | +6.75 | +7.04 | +11.30 |
| Random search | +6.89 | +7.15 | +12.04 |
| Disease grouping | +6.29 | +6.99 | +8.61 |

Specifically, we replace progressive sampling and greedy search with purely random methods, referred to as Random Sampling and Random Search. In all three settings, performance generally decreases, highlighting the contributions of these components of `AutoDP`.

Additionally, we use disease-based grouping (Appendix B) to first assign tasks into different groups based on their medical relevance and then employ DARTS to search for the model architecture for each group. This allows us to analyze the effectiveness of automated task grouping. By comparing disease-based grouping with the searched configurations (Appendix C), we observe that `AutoDP` does not strictly follow medical classifications for task grouping but achieves significant performance improvements over disease-based grouping. This indicates the necessity of using an automated search algorithm to find the optimal task grouping, which surpasses human intuition.

## 5 Conclusions and Future Work

In this paper, we propose `AutoDP`, an automated multi-task learning framework for joint disease prediction on EHR data. Compared to existing work, our method largely improves the design of task grouping and model architectures by reducing human interventions. Experimental results on real-world EHR data demonstrate that the proposed framework achieves significant improvement over existing state-of-the-art methods, while maintaining a feasible search cost. There are also some valuable future directions based on the current version of `AutoDP`.

First, from the application perspective, if we aim at deploying `AutoDP` to real-world healthcare systems, it would be advantageous to apply it to more complex problem settings. For example, the incorporation of diverse clinical data sources beyond EHR such as claims, drugs, medical images and texts will significantly enhance the practical utility of `AutoDP`.

Additionally, considering the dynamic nature of healthcare environments with continuously updated input data and evolving tasks, adapting the surrogate model to accommodate new data and tasks would be imperative.

Moreover, addressing privacy concerns within healthcare systems is a promising direction. Therefore, extending `AutoDP` with data processing pipelines for automatic feature engineering could offer enhanced privacy safeguards and further improve its applicability in sensitive healthcare contexts.

Finally, we assume all the tasks have the same input EHR data in our problem setting, which might not always be the case in practical scenarios. Chances are that, for some diseases, there are large and well-annotated data, while for the others, there are limited data available. How we should extend the current framework to handle more heterogeneous diseases/tasks remains a challenge.

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

## Appendix

We include the following sections in Appendix, providing the additional details of the proposed framework, additional experimental results, and a discussion about limitation and future work.

## A  Implementation

To prepare our dataset, we adopt the data pre-processing pipeline outlined in Harutyunyan, et al. [27]. Given that the original implementation[2] is designed for MIMIC-III [33], we make specific modifications to tailor it for MIMIC-IV. The 25 labels are defined using the Clinical Classifications Software (CCS) for ICD-9 code[3]. Consequently, we first map the ICD-10 codes[4] in the MIMIC-IV database to ICD-9 codes before generating the labels. After processing, we have the feature dimension $d_e$ as 76. We partition the dataset as train, validation and test sets with a ratio of $0.7 : 0.15 : 0.15$.

We implement the framework using the PyTorch framework and run it on an NVIDIA A100 GPU. Given the dataset we have, we first train a vanilla LSTM for every task independently, and report the backbone performance in Table 3, which can be further used to compute multi-task gains. For the proposed method, we run three settings of experiments: **Task @ 5**, **Task @ 10** and **Task @ 25**, which refers to using the first 5 tasks, 10 tasks and 25 tasks respectively. For different settings, we use specific hyperparameters as shown in Table 4. Besides that, we define the candidate operation set $\mathcal{O}$ as {Identity, Zero, FFN, RNN , Attention}, which includes widely used operations for processing EHR time series. Among them, Identity means maintaining the output identical to the input. Zero means setting all the values of the input feature to 0. Attention and FFN represents one self-attention layer and one feed-forward layer respectively, which are the same as in Transformer [30]. RNN is one recurrent layer, and we adopt LSTM [32] in our framework. For all the MTL procedures and baseline training, we apply the batch size of 64 and learning rate of $3e-4$. For training the surrogate model, we use the batch size of 5 and learning rate of $5e-5$. During searching, we compute all multi-task gains on the validation set for guiding the surrogate model training. After we obtain the optimal configuration, we train the searched models and report their multi-task gains on the test set.

## B  Disease Based Grouping

To show the effectiveness of automated task grouping, we conduct experiments using a predefined task grouping based on disease categories. We asked GPT-4 [34] to classify the 25 prediction tasks into different groups based on their medical meaning. The result is shown in Table 5. Using this grouping, we further apply DARTS to each group and report the multi-task gains as shown in Table 2. Compared to **AutoDP**, there is a notable performance drop for the disease based grouping. This means human intuition dose not provide the optimal task grouping, which underscores the necessity of employing search algorithm to automatically discover better task grouping for MTL.

## C  Visualization of the Searched Configurations

Here, we show two example of the final configuration for setting **Task @ 10** in Figure 4 and for **Task @ 25** in Figure 5. The proposed **AutoDP** identifies 5 and 10 different task groups respectively and also searches for the corresponding architectures. We can observe that some of the tasks tend to be trained independently, while others are grouped together for joint training. This supports our claim that fine-grained task grouping is necessary to bring the optimal performance gain. Also, the optimal architecture is also different for each task group, which further justifies the necessity of joint optimization over task grouping and architecture search.

---

[2]https://github.com/YerevaNN/mimic3-benchmarks
[3]https://www.cdc.gov/nchs/icd/icd9.htm
[4]https://www.cms.gov/medicare/coding-billing/icd-10-codes/
2018-icd-10-cm-gem

Table 3: Performance of the single task backbone.

| Task | ROC | AVP |
|---|---|---|
| Acute and unspecified renal failure | 0.7827 | 0.5647 |
| Acute cerebrovascular disease | 0.9079 | 0.4578 |
| Acute myocardial infarction | 0.7226 | 0.1761 |
| Cardiac dysrhythmias | 0.6948 | 0.5168 |
| Chronic kidney disease | 0.7296 | 0.4383 |
| Chronic obstructive pulmonary disease and bronchiectasis | 0.6791 | 0.2689 |
| Complications of surgical procedures or medical care | 0.7229 | 0.4045 |
| Conduction disorders | 0.6712 | 0.1880 |
| Congestive heart failure; nonhypertensive | 0.7601 | 0.5129 |
| Coronary atherosclerosis and other heart disease | 0.7351 | 0.5589 |
| Diabetes mellitus with complications | 0.8844 | 0.5559 |
| Diabetes mellitus without complication | 0.7484 | 0.3355 |
| Disorders of lipid metabolism | 0.6730 | 0.5816 |
| Essential hypertension | 0.6298 | 0.5258 |
| Fluid and electrolyte disorders | 0.7396 | 0.6129 |
| Gastrointestinal hemorrhage | 0.7076 | 0.1281 |
| Hypertension with complications and secondary hypertension | 0.7141 | 0.4243 |
| Other liver diseases | 0.6849 | 0.2303 |
| Other lower respiratory disease | 0.6371 | 0.1417 |
| Other upper respiratory disease | 0.7602 | 0.2228 |
| Pleurisy; pneumothorax; pulmonary collapse | 0.7051 | 0.1417 |
| Pneumonia | 0.8171 | 0.3786 |
| Respiratory failure; insufficiency; arrest (adult) | 0.8651 | 0.5497 |
| Septicemia (except in labor) | 0.8291 | 0.4866 |
| Shock | 0.8792 | 0.5574 |

Table 4: Hyperparameter setting.

| Parameters | | Task @ 5 | Task @ 10 | Task @ 25 |
|---|---|---|---|---|
| # of tasks | $N$ | 5 | 10 | 25 |
| Dimension of $F(\cdot)$ | $d_s$ | 64 | 64 | 64 |
| # of nodes | $P$ | 2 | 2 | 3 |
| Progressive sampling | $Q_0$ | 10 | 10 | 20 |
| | $Q_1$ | 50 | 100 | 100 |
| | $Q_2$ | 10 | 20 | 20 |
| | $\lambda$ | 0.5 | 0.5 | 0.5 |
| | $K_1$ | 20 | 30 | 25 |
| Greedy search | $K_2$ | 1000 | 1000 | 1000 |
| | $B$ | 3 | 5 | 10 |
| Runtime | GPU Hours | $\sim 20$ | $\sim 75$ | $\sim 200$ |

Table 5: Disease Based Grouping.

| Groups | Diseases |
|---|---|
| Cardiovascular Diseases | Acute cerebrovascular disease
Acute myocardial infarction
Cardiac dysrhythmias
Congestive heart failure; nonhypertensive
Coronary atherosclerosis and other heart disease
Essential hypertension |
| Respiratory Diseases | Chronic obstructive pulmonary disease and bronchiectasis
Other lower respiratory disease
Other upper respiratory disease
Pleurisy; pneumothorax; pulmonary collapse
Pneumonia (except that caused by tuberculosis or sexually transmitted disease)
Respiratory failure; insufficiency; arrest (adult) |
| Kidney Diseases | Acute and unspecified renal failure
Chronic kidney disease |
| Metabolic Diseases | Diabetes mellitus with complications
Diabetes mellitus without complication
Disorders of lipid metabolism
Fluid and electrolyte disorders |
| Gastrointestinal Diseases | Gastrointestinal hemorrhage |
| Infections | Septicemia (except in labor) |
| Surgical/Medical Complications | Complications of surgical procedures or medical care |
| Neurological/Cardiac Conditions | Conduction disorders
Shock |
| Liver Diseases | Other liver diseases |

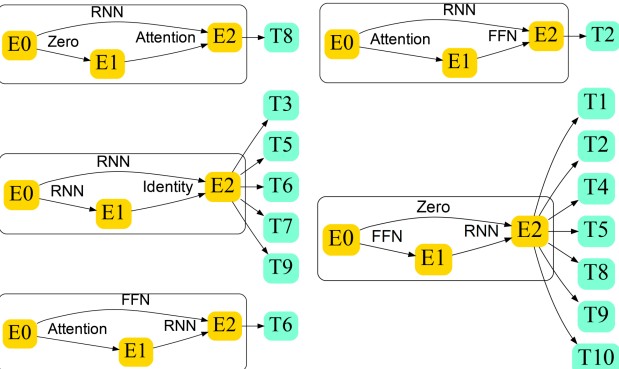

Figure 4: Illustration of the searched configuration under the setting of Task @ 10.

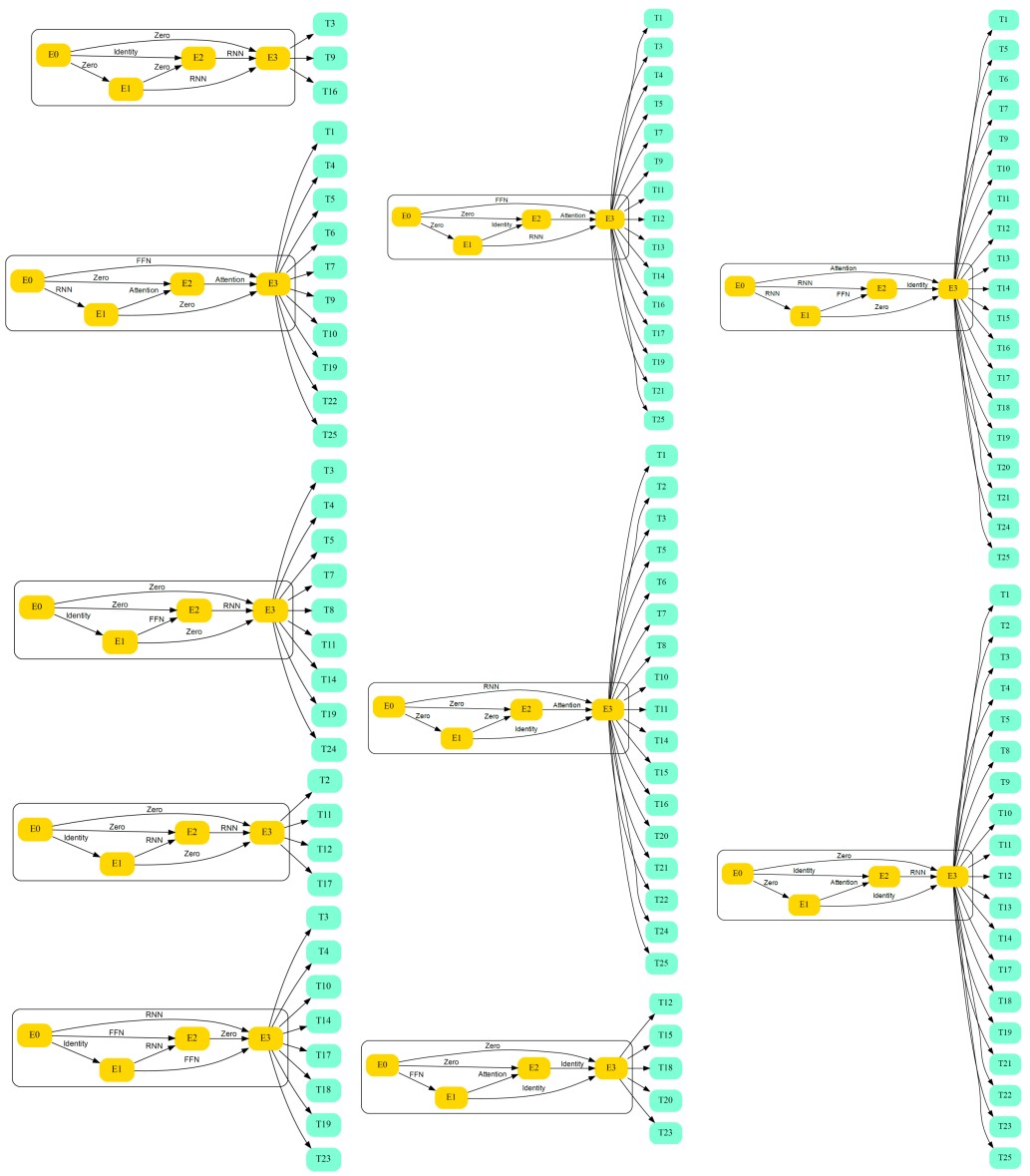

Figure 5: Illustration of the searched configuration under the setting of Task @ 25.

