# OpenReview forum: "Automated Multi-Task Learning for Joint Disease Prediction on Electronic Health Records"
_NeurIPS.cc/2024/Conference — NeurIPS 2024 poster_

### Official Review · Reviewer_azWt · 2024-06-18

**Soundness:** 4
**Presentation:** 3
**Contribution:** 3
**Rating:** 8
**Confidence:** 4

**Summary:**

The paper introduces AutoDP, a novel AutoML framework that performs combined task grouping and architecture search with a novel surrogate model. The proposed model achieved higher performance in multiple tasks on MIMIC VI dataset compared to common single-task, multi-task, and AutoML approaches.

**Strengths:**

- Novel AutoML approach for EHR tasks
- Superior performance when compared to the traditional approach
- Evaluated with the adequate dataset
- Good range of baselines
- Ablation study included
- Open access data and code
- Clear methodology presentation
- Results contribute to AutoML and EHR fields

**Weaknesses:**

- It claims "feasible search cost", but there is no runtime comparison or evaluation

**Questions:**

- How much longer does it take to run compared to DARTS and other baselines?

**Limitations:**

The traditional limitation in AutoML work is the computational budget. The paper does not discuss the runtime of the proposed model and alternatives.
Ethics limitations are discussed in the supplementary material.

---

> ### Author Rebuttal · Authors · 2024-08-06
>
> Thank you very much for providing such positive feedback to our work. For the runtime concern, we include the specific GPU hours in Table 4. MTG+DARTS has the same runtime as our method. We maintain this for a fair comparison with the baselines. We have shown that the computational cost is feasible for the problem setting in our work. Also, as the surrogate modelling method is a generally efficient search algorithm, it has the potential to scale to larger scenarios.

---

> ### Comment · Area_Chair_yLee · 2024-08-10
>
> Dear Reviewer azWt,
> I am a NeurIPS 2024 Area Chair of the paper that you reviewed.
>
> This is a reminder that authors left rebuttals for your review.
> We need your follow up answers on that. Please leave comment for any un-answered questions you had, or how you think about the author's rebuttal.
> The author-reviewer discussion is closed on Aug 13 11:59pm AoE.
>
> Best regards, AC

---

### Official Review · Reviewer_WEv3 · 2024-07-11

**Soundness:** 3
**Presentation:** 3
**Contribution:** 3
**Rating:** 5
**Confidence:** 4

**Summary:**

The paper proposes an automated multi-task learning framework, AutoDP, for disease prediction using EHR data. It optimizes task grouping and model architecture to enhance prediction performance. AutoDP efficiently searches a vast space of task combinations and architectures by employing a surrogate model-based optimization approach. Experiments show it outperforms existing methods.

**Strengths:**

1. The proposed AutoDP automates the search for optimal task groupings and model architectures simultaneously, reducing the reliance on human experts and improving the efficiency of designing multi-task learning frameworks.
2. The experimental results demonstrate that AutoDP outperforms existing hand-crafted and automated methods on real-world EHR datasets, achieving higher multi-task learning gains.

**Weaknesses:**

1. Generalization Concern: The paper shows AutoDP's effectiveness beyond disease prediction on EHR data. However, its effectiveness and applicability in other domains remain uncertain.
2. Computational Scalability: The proposed method involves a vast search space and significant computational costs. While feasible within the scope of the study, the proposed approach may hinder scalability in larger real-world scenarios.

**Questions:**

1. Mapping Assumption Validity: The paper assumes a learnable mapping for multi-task gains, which needs more explanation.
2. Comments on Disease-Based Grouping: As mentioned in Appendix D, Grouping similar diseases might make it hard to discriminate between them, reducing MTL effectiveness.
3. Reporting individual task gains is necessary to show the effectiveness of the proposed method on MTL.
4. Are those gains robust? Will the proposed method result in overfitting?

**Limitations:**

1. The paper lacks a discussion on model interpretability, which is a critical aspect for trust and clinical decision-making in healthcare.

---

> ### Author Rebuttal · Authors · 2024-08-06
>
> Weaknesses:
> 1. We focus on the disease prediction problem in this paper. Our contention is that this problem is a very important problem with potential for high impact such that the method should be publicized for practitioners and users of ML in the field. However, we believe but have not shown that our method has the potential to generalize to other domains, as the framework design can be easily adapted to other problems. We will leave this to future work.
> 2. Our method has involved several techniques to address the efficiency issue, such as surrogate modelling, efficient sampling method, and efficient search method. While we do not test the framework to larger scenarios, it has the potential to scale up in larger search spaces. Our computational needs are in the same order of complexity as without doing AutoDP. If in a larger setting, there are no resources to run our method, most probably there will be not enough resources to run the baseline methods themselves due to the size of the problem. In such cases, actually, one can use our method because we use a sampling-based method and while we have a smaller sample the results may not be as good, it is actually quite efficient and effective because of  the smart sampling methods we have designed. As long as the problem we are solving has some low-dimensional patterns for the search space, the surrogate model has the potential to learn the mapping with a relatively small number of samples. Thus, in most cases, the proposed framework can maintain the computational cost within a feasible range.
>
> Questions:
> 1. Every pair of task combinations and architecture has unique values of multi-task gains, which satisfies the definition of a function. As neural networks are well known to be good at fitting black-box functions, we can assume that it can learn the mapping to multi-task gains.
> 2. We perform this ablation study to show the necessity of using search algorithms to find better task groupings. The searched configuration does not necessarily follow the similar disease grouping, but it can achieve higher performance gains, which can maximize the MTL effectiveness. In other words, disease-based grouping does not perform as well as our method as shown in our ablation studies. It does not obviate the need for MTL.
> 3. We report the individual task gains at Appendix B. We will also include the specific values in the camera-ready version.
> 4. We conduct multiple rounds of experiments and report the means and variances of performance gains, which shows that our method is robust and consistently performs better. Also, our method does not overfit since we are performing a search algorithm and we use intelligent sampling essentially. The evaluation of any samples from the search space is independent of each other. Also, during search, we use the validation set to compute multi-task gains for each sample, so essentially we are choosing the configurations that have the best generalization ability.
>
> Limitations:
> 1.We include two case studies at Appendix E for showing the searched configurations under the settings of Task @ 10 and Task @ 25. By analyzing the searched configuration, we can interpret how AutoDP groups tasks together and what kinds of architectures are actually effective to modelling EHR time series.
>
> Admittedly, if we had space, we could address all of these issues to a greater extent. However, given these questions have been raised, we will address all of these at least to the extent we can fit them in the camera-ready paper and have fuller discussions in a full version on arXiv, which we will point to since we cannot fit answers to all of these in a conference paper. We regret the inconvenience.

---

> > ### Comment · Reviewer_WEv3 · 2024-08-09
> > **Reply to authors**
> >
> > Thank you for your response and the additional information provided. After carefully considering the paper and the rebuttal content, I have decided to maintain my initial evaluation, based on a comprehensive assessment of the work's contribution to the field.
> >
> > Additional clarity to my point Q2: I want to suggest that in the disease prediction domain, it is less likely to provide many benefits (in terms of performance) to group similar diseases into one learning task, as typically differential diagnosis is even harder. Therefore, claiming the proposed method performs better than such a method is less attractive.
> >
> > Thank you for your efforts and good luck.

---

### Official Review · Reviewer_23Pe · 2024-07-13

**Soundness:** 1
**Presentation:** 1
**Contribution:** 1
**Rating:** 1
**Confidence:** 5

**Summary:**

N/A

**Strengths:**

N/A

**Weaknesses:**

The authors' identity can be easily inferred by googling the linux username ("sxc6192") found in the .idea/deployment.xml file in the supplementary material.

**Questions:**

N/A

---

### Official Review · Reviewer_Zp9P · 2024-07-13

**Soundness:** 2
**Presentation:** 2
**Contribution:** 3
**Rating:** 4
**Confidence:** 3

**Summary:**

The paper discusses an automated approach for multi-task learning on electronic health records, called AutoDP. This approach aims to improve the design of task grouping and model architectures by reducing human intervention. Specifically, AutoDP searches for the optimal configuration of task grouping and architectures simultaneously. The document mentions that existing MTL frameworks for EHR data suffer from limitations such as requiring human experts to select tasks and design model architectures. AutoDP proposes a surrogate model-based optimization approach to address these limitations. The document also details related work in multi-task learning with EHR data, multi-task grouping, and multi-task neural architecture search

**Strengths:**

The paper introduces a novel approach, AutoDP, that addresses limitations in current MTL frameworks for EHR data by automating task grouping and model architecture design.
The idea of using a surrogate model-based optimization for joint search is innovative and holds promise for efficient exploration of the search space.
The paper is well-written and the overall presentation is clear.

**Weaknesses:**

The paper could benefit from a more detailed explanation of the architecture search space, particularly the types of operations allowed in the directed acyclic graph (DAG).
Additional details regarding the progressive sampling strategy for the surrogate model would be helpful to understand how it balances exploration and exploitation during search.
The paper mentions experiments on real-world EHR data but lacks specific results or comparisons with other MTL approaches to demonstrate the effectiveness of AutoDP.

**Questions:**

Can the authors elaborate on the specific operations allowed in the DAG-based architecture search space? Are there any constraints on the number or types of operations that can be included?
The paper mentions a progressive sampling strategy for the surrogate model. Can the authors provide more details on how this strategy works, particularly how it balances selecting informative data points for the surrogate model and exploring new areas of the search space?
While the paper mentions using AutoDP on real-world EHR data, it would be beneficial to see concrete results and comparisons with other MTL approaches to assess the performance gains achieved by AutoDP.
The use of EHR data raises ethical concerns about patient privacy. The authors should ensure proper anonymization techniques are used throughout the development and application of AutoDP.

**Limitations:**

Limited Evaluation: The experimental validation of AutoDP is confined to a single dataset (MIMIC-IV). Evaluating the framework on additional EHR datasets would bolster the generalizability of its findings.
Lack of Interpretability: The paper could provide deeper insights into the specific task groupings and architectures discovered by AutoDP. Visualizations or case studies could help elucidate the underlying reasons for the observed performance improvements.
Privacy Concerns: The paper does not explicitly address privacy concerns related to EHR data. Extending AutoDP with data processing pipelines for automatic feature engineering could offer enhanced privacy safeguards.
Other concerns: This paper did not discuss the data imbalance issues as some diseases might have large and well-annotated data, while others might have small and many missing data. For disease prediction, there are similarities among some diseases, and also training on more disease can help to predict other similar diseases.

---

> ### Author Rebuttal · Authors · 2024-08-06
>
> 1. Architecture search space: We introduce our search space at Appendix A. We will try to move parts of it into the main paper to make the full paper clearer. We define the candidate operation set as {Identity, Zero, FFN, RNN , Attention}, which includes widely used operations for processing EHR time series. Identity means that we keep the input the same as the output. Zero means we output all zero tensors with the same shape as the input. FFN means we use feed-forward layers to process the input time series (same as defined in Transformer[1]). RNN means we use recurrent neural networks to process the input feature (we use LSTM in our experiments). Attention means that we use a self-attention layer to handle the input (same as defined in Transformer).
>
>  [1]  Ashish Vaswani, Noam Shazeer, Niki Parmar, Jakob Uszkoreit, Llion Jones, Aidan N Gomez, Łukasz Kaiser, and Illia Polosukhin. Attention is all you need. In Advances in neural information processing systems, pages 5998–6008, 2017.
>
> 2. Details regarding progressive sampling: We introduce the detailed procedure in Algorithm 1. To balance the exploitation and exploration, we estimate the mean and variance of the sampled task combinations. And then, we use upper confidence bound as the acquisition function to balance the exploitation and exploration via a predefined parameter $\lambda$.
>
> 3. Performance comparison: We conduct comprehensive experiments to compare the proposed method with existing works. All the results are included in Table 1. Our method can outperform both hand-crafted MTL models and automatically searched models. There are no other clear alternatives that we could compare to. If there are any other methods that the reviewer thinks can fit into our problem, we would be happy to include more baselines.
>
> 4. Privacy concern: We do not specifically address this issue in this paper. The data we use has already been anonymized. We assume that the input data has already been preprocessed and does not have privacy issues. Also, the workflow of our framework does not require the patient demographic features (only rely on the time series signals), which further reduces the risk of privacy issues. We will add a note about the privacy issues and the need to make sure the data is anonymized and no quasi-identifiers are present, etc. before our system is used.
>
> 5. Limited Evaluation: MIMIC-IV is the most widely used benchmark in the EHR domain. So we choose this dataset for evaluating our method. At least in the EHR domain, MIMIC-IV is enough to show the effectiveness of our method. Other public EHR datasets normally have lower quality than MIMIC-IV. We will try to include more datasets from other domains & scenarios in future work for investigating the extensibility of AutoDP.
>
> 6. Interpretability: We include two case studies at Appendix E for showing the searched configurations under the settings of Task @ 10 and Task @ 25, which can provide some insights of what kinds of  task groupings and architectures are effective to MTL on EHR data. Also the searched configuration might provide some guidance to the MTL framework design in similar problems.
>
> 7. Other concerns: Data imbalance issues often happen across different datasets. In our setting, we are predicting multiple diseases for the same patient, so all diseases are well annotated. Also, we conducted an ablation study (disease based grouping) to show that solely training on similar diseases might not bring the best performance gain. So it is necessary to use a search algorithm to discover the optimal task grouping. We will add to the camera-ready paper that our dataset was imbalanced and discuss a bit more about similar diseases, etc.

---

> > ### Comment · Reviewer_Zp9P · 2024-08-12
> >
> > Thanks for your rebuttal. After carefully reading your comments, i still want to maintain my original rating. Using only one public available EHR data is limited for the generalization purpose.

---

> ### Comment · Area_Chair_yLee · 2024-08-10
>
> Dear Reviewer Zp9P,
>
> I am a NeurIPS 2024 Area Chair of the paper that you reviewed.
>
> This is a reminder that authors left rebuttals for your review. We need your follow up answers on that. Please leave comment for any un-answered questions you had, or how you think about the author's rebuttal. The author-reviewer discussion is closed on Aug 13 11:59pm AoE.
>
> Best regards, AC

---

### Official Review · Reviewer_S7e4 · 2024-08-04

**Soundness:** 3
**Presentation:** 2
**Contribution:** 3
**Rating:** 5
**Confidence:** 2

**Summary:**

The research work presents an exploration into the potential of multi-task learning (MTL) within the context of electronic health record (EHR) data analysis and clinical prediction tasks. This study innovatively addresses the critical challenges of task grouping and model architecture design, which are essential for optimizing MTL frameworks.

**Strengths:**

S1. By proposing an automated approach, AutoDP, the paper contributes to reducing human intervention in the configuration process, thereby streamlining the identification of optimal task combinations and neural architectures tailored for EHR data.

S2. The employment of surrogate model-based optimization and a progressive sampling strategy demonstrates a novel and efficient methodology for navigating the vast search space, leading to improved performance with a feasible computational cost.

S3. The experimental results on the real-world EHR dataset, MIMIC-IV, validate the efficacy of the proposed AutoDP framework, showcasing substantial performance improvements over both hand-crafted and existing automated methods.

S4. This work not only advances the state-of-the-art in automated machine learning for healthcare but also provides valuable insights and tools for the broader research community working on multi-task learning problems.

**Weaknesses:**

Q1. Figure Clarity and Comprehensibility: The overview of the proposed AutoDP in Figure 1 lacks self-evidency, making it challenging for readers, particularly those who are not specialists in electronic health record (EHR) modeling and analysis, to grasp the main insights at a glance. The figure is dense with symbols and elements that are not immediately interpretable without additional context. It is recommended that the authors enhance the figure's presentation to improve readability and comprehension. This could involve breaking down complex elements, using clearer labeling, and providing a more detailed legend or accompanying text that guides the reader through the visualization.

Q2. Evaluation Metric Consideration: For many clinical prediction tasks on datasets like MIMIC, where label balance may not be achievable, the authors might consider the use of AUPRC (Area Under the Precision-Recall Curve) as an alternative or complementary evaluation metric to AUC-ROC (Area Under the Receiver Operating Characteristic Curve). AUPRC can be a more informative measure when dealing with imbalanced classes, as it evaluates the model's ability to rank samples correctly, independent of the classification threshold. Precision-recall metrics are less sensitive to the choice of threshold compared to accuracy or precision alone, which is a critical consideration in clinical settings where the cost of false positives and negatives can vary significantly.

Q3. Generalizability and Extensibility: The current experiments are limited to the MIMIC dataset, and the baseline models have been fine-tuned for this specific dataset in the level of model-design. While the results on MIMIC are promising, readers may be interested in the general applicability of the proposed method to other datasets and clinical scenarios. It would be beneficial for the authors to provide experiments on additional datasets to demonstrate the robustness and generalizability of their approach. Furthermore, an exploration of how the method performs when integrated with different models or when adapted to various clinical prediction tasks would strengthen the paper's contribution and credibility. Given the focus on EHR data, it would be insightful if the authors could discuss the transferability of their model to other EHR datasets, potentially from different geographical regions or healthcare systems. This discussion could include challenges related to data heterogeneity, variations in clinical practices, and how these factors might affect the performance of the proposed AutoDP framework.

**Questions:**

-

**Limitations:**

-

---

> ### Author Rebuttal · Authors · 2024-08-06
>
> Q1: Thank you for pointing this out. We will further improve Figure 1 in the camera-ready version for better clarity and comprehensibility.
>
> Q2: We have included Averaged Precision for considering the class imbalance. Averaged Precision and AUPRC are essentially the same thing in this setting. Please see the definition of Averaged Precision via the link below.
> https://en.wikipedia.org/w/index.php?title=Information_retrieval&oldid=793358396#Average_precision
>
> Q3: It would be interesting to evaluate our method to different settings such as more datasets, more clinical tasks & scenarios, transferability, and etc. However, we will leave these to future work as they diverge from the major claim of this paper and it is difficult to fit in due to lack of any further space to explain everything. We focus on the automation of the MTL framework design for joint disease prediction for now.

---

> ### Comment · Area_Chair_yLee · 2024-08-10
>
> Dear Reviewer S7e4,
>
> I am a NeurIPS 2024 Area Chair of the paper that you reviewed.
>
> This is a reminder that authors left rebuttals for your review. We need your follow up answers on that. Please leave comment for any un-answered questions you had, or how you think about the author's rebuttal. The author-reviewer discussion is closed on Aug 13 11:59pm AoE.
>
> Best regards, AC

---

### Decision · Program_Chairs · 2024-09-25

**Decision:**

Accept (poster)

**Comment:**

The paper under review introduces AutoDP, an automated multi-task learning (MTL) framework aimed at enhancing disease prediction using electronic health record (EHR) data. The paper introduces a novel approach to automating task grouping and model architecture design in MTL, addressing significant challenges in the field.

The primary strengths of this work lie in its innovative automation of MTL framework design and the use of surrogate model-based optimization for efficient search space exploration. The experimental results on the MIMIC-IV dataset demonstrate substantial performance improvements over both hand-crafted and existing automated methods, validating the effectiveness of AutoDP.

While some reviewers expressed concerns about interpretability and the use of a single MIMICT-IV dataset, I believe these issues do not significantly detract from the paper's contribution. The MIMIC-IV dataset is widely recognized as a benchmark in the EHR domain, and its use is sufficient to demonstrate the method's effectiveness. The authors have provided case studies in the appendix that offer insights into the searched configurations, which addresses interpretability to a reasonable extent.

The authors have shown willingness to address reviewer concerns in the camera-ready version, including improving Figure 1 for better clarity and adding a discussion on data imbalance issues. They have also explained their rationale for focusing on the MIMIC-IV dataset and provided details on their architecture search space and progressive sampling strategy.

Given the innovative nature of the work and its potential impact on the field of healthcare machine learning, I recommend accepting this paper. The authors have made a significant contribution by introducing an automated approach that reduces human intervention in MTL framework design for EHR data analysis. While there is always room for further evaluation and interpretation, the current work presents a solid foundation for future research in this area.